# A Preliminary Study of SARS-CoV-2’s Permanence and Potential Infective Capacity in Mineromedicinal Waters of Copahue, Neuquén, Argentina

**DOI:** 10.3390/ijerph19105923

**Published:** 2022-05-13

**Authors:** María Lorena Vela, Gisela Masachessi, María Alejandra Giaveno, Maria Eugenia Roca Jalil, Gonzalo Castro, Ariana Mariela Cachi, María de los Ángeles Marinzalda, Ana Zugarramurdi, Miria Baschini

**Affiliations:** 1Health and Environment Sciences School, Comahue National University, Neuquen 8300, Argentina; 2Neuquén Provincial Thermal Organization (E.Pro.Te.N.), Neuquen 8349, Argentina; anazugarra@hotmail.com; 3Viral Gastroenteritis and Measles Laboratory, “Dr J. M. Vanella” Virology Institute, Health Science School, Córdoba National University, Córdoba 5000, Argentina; gmasachessi@fcm.unc.edu.ar; 4National Council for Scientific and Technical Research (CONICET), Buenos Aires 1425, Argentina; 5Engineering School, Comahue National University, Neuquen 8300, Argentina; alejandra.giaveno@fain.uncoma.edu.ar (M.A.G.); eugenia.rocajalil@probien.gob.ar (M.E.R.J.); miria.baschini@fain.uncoma.edu.ar (M.B.); 6Institute for Research and Development in Process Engineering, Biotechnology and Alternative Energies (PROBIEN), CONICET-Comahue National University, Neuquen 8300, Argentina; 7Central Laboratory Department, Ministry of Health of the Province of Córdoba, Córdoba 5000, Argentina; gonmcastro@gmail.com; 8National Institute of Aeronautical and Space Medicine, FAA, Córdoba 5000, Argentina; arianamcachi@gmail.com (A.M.C.); m.marinzalda@hotmail.com (M.d.l.Á.M.); 9Faculty of the Air Force, National Defense University, Córdoba 5000, Argentina

**Keywords:** Copahue, SARS-CoV-2, spring water, survival, pandemic

## Abstract

Copahue Thermal Center is characterized by the presence of mineromedicinal acidic waters with high temperatures, therapeutic peloids, and relevant consortia of extremophiles species, distributed in small natural pools which cannot be disinfected. The objective of this research was to investigate the survival of SARS-CoV-2 in Copahue’s waters and its remaining infective capacity. In a first assay, a decrease of more than 50% of the initially viral load compared to the initially inoculated positive sample was detected for all the water samples analyzed. After that, two of the Copahue springs, which are used as an immersion bath in closed environments without going through any disinfection treatment, was selected to determine the viral viability. VERO cell infections were performed, with no cytopathic effect detected, but a strikingly high resistance of the virus, detecting its genome by real time PCR, during the seven days of study under laboratory conditions. SARS-CoV-2 survival in acid media was reaffirmed, which is a peculiarity for a covered virus. A decrease in the detectable viral load of the positive sample was found as the infection time passed, becoming completely negative in the subsequent blind passages. More research is needed to further study the feasibility of SARS-CoV-2 in mineromedicinal waters, especially natural acidic waters that cannot disinfected, in order to expand information about the risk to populations that are exposed to them.

## 1. Introduction

The years 2020 and 2021 were hit by the highly contagious Coronavirus SARS-CoV-2 pandemic, which initially caused the death of people over the age of 60 and people with comorbidities. The various thermal centers around the world, where mineromedicinal waters constitute a main resource, were initially forced to close their doors and, at a later stage, had to elaborate detailed protocols for their reopening which could guarantee patient safety and avoid contagion [1].

At the beginning of the pandemic, many aspects related to virus transmission were rather unclear, making it necessary to delve into various experimental studies on human contagion, especially in relation to aquatic media [2]. Strict disinfection and water chlorination protocols were followed in many sites, which proved to be successful in reducing the contagiousness of such systems by at least three orders of magnitude, hence allowing for the use of the community swimming pools [3].

In South America, especially in natural thermal waters where the biological diversity is still present due to the ecosystem’s unadulterated condition [4,5,6,7], disinfection protocols by chlorination or through the use of any other disinfectant would be altogether unsuitable as it would result in the death of natural microorganisms, which are vital for these waters’ ultimate pharmacological activity. Moreover, in Copahue, most of the waters are naturally acidic [8,9,10], which is a relevant additional reason to avoid any kind of disinfection treatment.

Additionally, the baths have served through history as key health aid centers, not only for seasonal sickness recovery but also for patients affected with other globally-spread contagious diseases [11], so proper knowledge about the secure usage of the baths is imperative during pandemic times.

At thermal centers, water used by a large number of people generally requires routine measurements in quality control and to that effect water recirculation, considering its physicochemical parameters, are a common practice [12,13].

The Copahue Thermal Center is located in the Province of Neuquén, Argentina, nestled in the Andes Mountain range, at the bottom of the active volcano that bears the same name. It is characterized by the presence of acidic waters [14,15,16], high temperatures, the abundance of extremophile microbiota [17,18,19], and for aspects associated to the formation of quite unique peloids [8,9], the latter being part of treatments of patients who attend each year.

In a property of roughly 12 hectares surrounded by rock formations, (geolocation coordinates 37°51′08″ S 71°10′03″ O), we find the Copahue volcanic springs, with multiple volcanic vents, pothole-like formations, baths, and a series of lagoons containing waters, peloids, and gases that have been in use for therapeutic purposes since ancient times. This thermal center is known internationally for the quality of its natural resources which allows for the availability of a variety of health and wellness treatments in the complex, including physiotherapy, massages, whirlpool baths, mud therapy, and even access to a gymnasium.

A regulatory framework governs and protects the use and sustainability of these resources. The 1762 Act, sanctioned in 1988, by which the Thermal Neuquén Province Organization (or *E. Pro. Te. N.*) is created, establishing among its objectives the administration, promotion, protection, control, use, and exploitation of the mineromedicinal springs and the therapeutic muds under provincial domain and jurisdiction, as well as tending to the preservation of public health, through a rational use of the natural hydrothermal therapeutic resources, to attend to the sectors of the population in need.

The thermal center hosts a Balneotherapy complex called *Rolando Guevara*, consisting of a central nave with different sectors, natural lagoons, and a series of buildings where multiple balneotherapy techniques are used. The complex has a total area of 10,500 m^2^, of which 6000 m^2^ are built-up areas.

Up to 2600 daily thermal services are performed at the complex in different locations: 22 mud therapy boxes, 12 immersion pools, 11 steam booths, 1 gymnasium, 1 Hubbard tank, 3 physiotherapy boxes, 7 massage therapy boxes, as well as 5 medical offices and 2 nursing stations.

The importance of fully complying with the Act became imperative during the pandemic, so it was necessary to determine safety conditions of the thermal resources in times of health emergency, specifically regarding virus survival and their infective capacity in thermal environments. The Copahue Thermal Center has a provision of springs and lagoons, as can be observed in Figure 1.

There are three main lagoons, namely the “*Laguna del Chancho*” (Chancho’s Lagoon) which has sulfated water and its muddy bottom can be used for mud baths, the “*Laguna Sulfurosa*” (Sulfurous Lagoon) with its characteristic steam plume rich in H_2_S gas, and the “*Laguna Verde*” (Green Lagoon) with its impressive green color given by the micronization of Chlorella Kessleri algae.

At the surroundings there are several springs, each with its own name and its mineralizations, varying greatly from one another even though they are physically very close. The geological faults along Copahue give way to this peculiarity. These springs include “*Agua de Mate*” (Mate Water), and “*Agua Sulfurosa*” (sulfurous water), which are used in atmiatry techniques with nebulization, acting directly on the respiratory mucous membrane. “*Agua Baño 7/8*” (sulfurous bath 7/8 water) is also worth mentioning, characterized by its weakly mineralized water that is rich in H_2_S gases, and its five cabins where every year thousands of patients get immersed individually, to undergo specific treatments. It is important to state that the “*Agua de Volcan*” (volcano water) comes directly from the Copahue Volcano Crater it is collected using appropriate pipes, then it is stored in reservoirs and distributed for use in bathtubs.

In addition, the volcanic activity in the area results in the emission of different gaseous compounds [20], including hydrogen sulfide, which has been proven to potentially decrease the chances of the virus entering a healthy organism, when it is applied below toxic levels through aerosols distributed in the air and it accesses the respiratory tract [21]. Hot springs have been shown to be effective in patient recovery treatments post-COVID [22,23,24], which is yet another reason why secure handling of Copahue’s thermal resources should be evaluated.

As a large number of senior patients frequently visit Copahue each season, and considering the small size of its natural pools, with waters being reused in enclosed spaces, the scientific community working with natural resources of the Thermal Organization of the Province of Neuquén was put on alert. Moreover, the publication of research work indicating the detection of SARS-CoV-2 in sewage water, made us inquire about the survival of this virus in Copahue waters and its remaining infective capacity. The relevance of this conundrum lays in the fact that the acidic Copahue system allows for some human biosafety against pathogenic bacteria, but this might not be possible in the case of these viral particles that are capable of resisting acidic media, such as the one present in the stomach.

The aim of this study was to carry out a series of preliminary studies to verify if the SARS-CoV-2 virus had any possibility of remaining in the water system and maintaining its infective capacity, considering the complexity of Copahue waters from a physicochemical and microbiological point of view.

## 2. Materials and Methods

### 2.1. Mineromedicinal Water Analysis

The temperature and the redox potential were measured in situ. The pH and the conductivity of the mineromedicinal waters were measured after centrifugation in the laboratory, and their physical-chemical compositions were determined in accordance with the Standard Methods for the Examination of Water and Wastewater [25].

### 2.2. Determination of Virus Permanence: Test I

Determination of the remaining viral load: Seven water samples were selected from different thermal springs which were labeled: Chancho’s Lagoon, Green Lagoon, Sulfurous Lagoon, Mate Water, Sulfurous Water, Sulfurous Bath 7/8 Water and Volcano Water, and were later transferred to the Neuquén Province Central Laboratory. Three aliquots of each sample were infected with a clinical sample of nasopharyngeal swab which had been previously diagnosed positive for SARS-CoV-2 with a viral load of 5754 viral copies/µL and they were let to incubate for three hours on a rotary agitator. Afterwards, the appropriate nucleic acids extraction was performed and the viral DNA was quantified through qPCR technique, expressing the results as an obtained data average for each sample [26,27].

### 2.3. Determination of SARS-CoV-2’s Infective Capacity: Test II

Two water samples were selected according to acidity, mineralization, and temperature, as well as other criteria related to their reutilization, their use in enclosed spaces, and the lack of chlorination systems. These samples were Green Lagoon and Sulfurous Bath 7/8 Water. The first one was used for immersion by large groups of patients and was later collected to be used in bathtubs without having been previously disinfected; the second analyzed source, the pools were used on demand, as immersion baths in enclosed areas without any disinfection treatment. Faced with the potential risk regarding the permanence of viral particles in the environment and in the water, the study was carried out on the potential remaining infective capacity these media.

Vero Cl76 cell-line (ATCC CRL-587) was used for virus isolation assays. The following protocol was performed: 250 µL from a positive SARS-CoV-2 swab, previously tested by cell culture to verify the viral viability of SARS-CoV-2, were added to a 250 mL aliquot of each mineromedicinal water sample. Later, a SARS-CoV-2 concentration was held for each sample [26,27]. The methodology is standardized for the concentration of enteric viruses from aqueous matrices, tuned for the concentration of SARS-CoV-2 [28]. The contaminated thermal waters with the virus suspension were briefly centrifuged at 4750× *g* during 20 min at 4 °C. The supernatant (S1) was kept at 4 °C to be used afterwards, and the sediment was mixed with a beef extract eluent at 3%/NaNO_3_ 2 M (pH 5.5) and agitated during 1 h at 4 °C. Then, the solids were eliminated by centrifugation at 10,000× *g* for 20 min and the eluate was mixed with the first supernatant obtained (S1) and pH was adjusted to 7.2. Polyethylene glycol (PEG) 6000 and NaCl were added at a 10% (*m*/*v*) and a 2% (*m*/*v*) final concentration, respectively. The resulting suspension was stirred overnight at 4 °C and centrifuged at 10,000× *g* for 25 min. The supernatant containing PEG was discarded, the sediment was suspended in a phosphate buffered saline (PBS) 2.5 mL (pH 7.2), adjusting pH to 8.0, it was incubated for 1 h at an occasional vortex and centrifuged at 10,000× *g* for 20 min. The supernatant was stored at −70 °C.

Afterwards, the viral concentrates were filtered using a 0.2 μm filter. Then, penicillin/streptomycin was added, and they were inoculated in Layton tubes containing a VERO Cell Line with their respective medium. The infection was performed using 150 μL of each sample. It was incubated at 37 °C for 1 h with soft agitation every 15 min. The cells were observed through the microscope to discard any presence of toxicity. Later, 1400 uL of Dulbecco’s modified Eagle’s medium (DMEM) supplemented with 2% fetal bovine serum (FBS) plus penicillin–streptomycin was added. It was incubated on a stove at 37 °C with 5% CO_2_. From that moment on, daily aliquots were taken to perform a PCR molecular technique in real time, having previously conducted genomic RNA extraction from each aliquot through the nucleoid acids automated extraction with a *GenePure* Pro Nucleic Acid Purification System NPA-32P–BIOER kit. Afterwards, a SARS-CoV-2 genome detection was performed using the *DisCoVery* SARS-CoV-2 RT-PCR Detection Kit (Safecare Biotech Hangzhou Co. Ltd., Hangzhou, China). The changes on the cycling threshold (Ct) value were measured. The Ct value can indicate the viral RNA load level in a sample (i.e., low Ct values correspond to high values of viral RNA concentrations, provided there are cells infected with SARS-CoV-2).

The genomic detection was accompanied by a daily observation of Layton tubes with an optic microscope to monitor the characteristic cytopathic effect from a SARS-CoV-2 infection. On the 7th day a VERO Cells monolayer blind passage was carried out with a 24-wells plate and from the 5th to the 7th day viral RNA detection was performed in real time through the RT-PCR technique.

## 3. Results and Discussion

### 3.1. Copahue, Mineromedicinal Waters and Biology

Considering the evaluated parameters showed in Table 1, the main characteristics of the waters are the following: Chancho’s Lagoon, sulfated, calcic, ferruginous; Green Lagoon, sulfated; Sulfurous Lagoon, sulphated; Sulfurous Bath 7/8 Water, sulfurous with predominant ions such as bicarbonates, calcium, sodium and magnesium; Mate water, predominant bicarbonate, calcium, sodium and sulfate ions; Sulfureous water, predominant bicarbonate, calcium, sodium and magnesium ions. Acidic waters prevail, the so-called Volcano water being especially acidic. As regards temperatures, they are generally hyperthermal, with Green Lagoon being the one with the lowest temperature value, which allows direct immersion of patients, without the need for additional procedures.

In addition to all the physical and chemical properties that mineromedicinal waters possess, they have a remarkable biological wealth of their own. The impossibility of chlorination to disinfect the waters of Copahue to achieve biological safety in the face of the pandemic caused by SARS-CoV-2, is not only related to its physicochemical properties but also to the microbiotic richness of the system. This ecological niche has been studied for more than 20 years using classical microbiology techniques as well as modern molecular biology methods, and many microbial species that colonize the different sites throughout the complex have been identified. Preponderance of archaea over thermophilic bacteria was detected in sites with higher temperature (>50 °C), whereas in water and mud with lower temperature (<50 °C) the predominant microorganisms were mesophilic and moderate thermophilic bacteria. In all cases, aerobic and anaerobic microorganisms were found depending on the amounts of oxygen present in each analyzed site. Regarding the physiological characteristics of the microorganisms, a great variety of metabolic capabilities present in both autotrophic and heterotrophic microorganisms was observed. The most notable were the capabilities to oxidize or reduce iron and sulfur, depending on the redox potential of each site that give it special characteristics to the system and that are responsible for the generation of acid in much of it. The prokaryotic community of the “*Volcán Copahue-Río Agrio*” (Copahue volcano-Sour River) system is formed mainly by acid bacteria and archaea, such as Acidithiobacillus, Ferroplasma, and Leptospirillum, which have been isolated from similar environments around the world. These results support the idea of a ubiquitous acidic biodiversity. However, this extreme environment that is of great interest apparently also hosts indigenous species such as *Sulfuriferula*, *Acidianus copahuensis*, and strains of *Acidibacillus* and *Alicyclobacillus*. Bacteria and archaea capable of oxidizing ammonium and nitrites have also been found actively participating in the nitrogen cycle. Some of the identified genera were Nitrospira Nitrolancea and Nitrosomonas, among others [29]. Additionally, several autochthonous species that colonize the Copahue geothermal park have been detected, among which a species of the genus Acidianus named A. copahuensis stands out. This species has a high metabolic variability characterized as acidophilic, thermophilic, facultative anaerobic, sulfur and iron oxidant [30]. Enrichment and isolation of acid-tolerant sulfate-reducing microorganisms was also performed in the anoxic and acidic thermal sediments of the Copahue volcano [31,32].

Ultimately, these geothermal waters house diverse and interesting consortiums of microorganisms adapted to extreme conditions, making them a genetic and biological diversity reserve that requires a great commitment in terms of its preservation.

Faced with the impossibility of chlorination the mineromedicinal waters of Copahue and considering the published data on the permanence of the virus in wastewater, as well as its ability to pass through acidic systems such as the stomach and remain intact [33,34], it was decided to carry out a series of preliminary tests to verify if the SARS-CoV-2 virus had any chance of remaining and maintaining its infective capacity after entering the system. For this purpose, trials I and II were designed and described in Section 2 (the results of which will be described below).

### 3.2. Virus Permanence

In test I, the inoculated viral load was 5754 viral copies/µL and after three hours viral DNA was still detected. The remaining percentages of viral load after the incubation period are shown in Table 2. A decrease of more than 50% of the initially inoculated viral load compared to the initially inoculated positive sample was detected for all samples. The remaining viral load was distributed in ranges, resulting in four groups, as shown in Table 2. The water from the volcano, which is the sample with the highest conductivity and lowest pH, was the one with the lowest viral load after incubation, while other samples such as those from group A were less aggressive to viral particles. However, it was not possible to obtain a pattern of variation in direct relation to the different physicochemical characteristics of the waters treated in this trial.

In Table 2 shows six mineral medicinal water, our intention was determined remaining viral load after three hours of incubation of the waters with an inoculum of SARS-CoV-2 virus in the seven spring water. For operational reasons, it was not possible to carry out further tests on the water of the Chancho’s Lagoon, one of the main lagoons of the Copahue hot springs. However, its exclusive outdoor use makes infective capacity determinations less relevant.

With the available methodology used it was also not possible to relate this remaining viral load with viral viability and therefore with the decrease in the infective capacity of the samples.

### 3.3. Infective Capacity

To meet this objective, select two of the Copahue springs already described were selected: the first one was the water from the Green Lagoon, as it was used for immersion by large groups of patients, and was later collected to be used in bathtubs without having been previously disinfected; the second analyzed source was the Sulfurous Bath 7/8 Water, as the pools were used on demand, as immersion baths in enclosed areas without any disinfection treatment. Faced with the potential risk regarding the permanence of viral particles in the environment and in the water, a study was carried out on the potential remaining infective capacity these media. Figure 2 shows the results obtained with the real time PCR technique for each assigned time, from 1 h as the first reading, up to one week (168 h). The evaluation was made by measuring changes in the Ct value.

The Ct value can indicate the relative load of viral RNA in a sample (lower Ct values imply higher viral levels, provided there is SARS-CoV-2 infection of cells).

Blind passages were negative for molecular detection of SARS-CoV-2. Genomic detection was accompanied by daily visualization of all assays (Layton tubes and 24-well plates containing blind passages) under an optical microscope to observe the characteristic cytopathic effect of SARS-CoV-2 infection. No cytopathic effect was detected in any of the tests carried out, as shown in Figure 3.

It is well known that the microbiological contamination of surface water courses can potentially make these water resources an important source of dissemination of viral agents and of viral infection risk for people exposed to these waters [35,36]. The SARS-CoV-2 pandemic continues to raise various questions about its infective potential in the different watercourses where it is found, particularly in freshwater sources that are affected by poorly treated or untreated wastewater, and in those water resources that serve as key health spaces for the recovery of patients affected by diseases [11]. 

Detection studies of the SARS-CoV-2 genome through infected cell cultures provide evidence of viable virus with infective capacity in the water source studied. To our knowledge, this is the first report that investigates whether mineromedicinal waters can be a source capable of sustaining the viability of SARS-CoV-2. The loss of genomic load, measured through the increase in Ct values, in the cell cultures studied and the non-detection of the SARS-CoV-2 genome in the subsequent blind passages, allows us to qualify the aqueous matrix studied as a source incapable of sustaining the infective potential of SARS-CoV-2. Therefore, we can infer that mineromedicinal waters would not be a potential source of viral infection in the exposed population.

## 4. Conclusions

The acidic mineromedicinal natural water of Copahue cannot disinfected to eliminate the SAR-CoV-2 virus due to both their physicochemical composition and their microbiological richness. 

There is an appreciable decrease in the detectable viral load of the positive sample when inoculated into the six samples from the Copahue hot springs used in this study, thus confirming a worrying feature of the new Coronavirus is its resistance to acidic media.

In order to relate this remaining viral load to viral viability, VERO cell infections were performed, with no cytopathic effect detected in Green Lagoon and Sulfurous Bath 7/8 Water, but a strikingly high resistance of the virus during the seven days of study under laboratory conditions.

A decrease in the detectable viral load of the positive sample was found in Green Lagoon and Sulfurous Bath 7/8 Water as the infection time passed, and the sample become completely negative in the subsequent blind passages.

Copahue mineromedicinal waters from Green Lagoon and Sulfurous Bath 7/8 Water would not be an adequate matrix for sustaining the viability of SARS-CoV-2.

More research is needed to deepen the studies on SARS-CoV-2′s viability in surface waters, to continue providing data on the risk for the population that is exposed to waters that potentially have presence of SARS-CoV-2.

## Figures and Tables

**Figure 1 ijerph-19-05923-f001:**
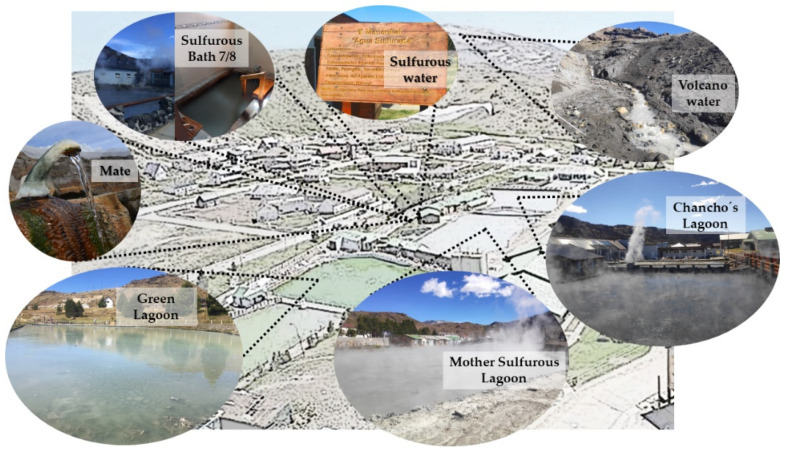
Spring allocation of the Copahue thermal system.

**Figure 2 ijerph-19-05923-f002:**
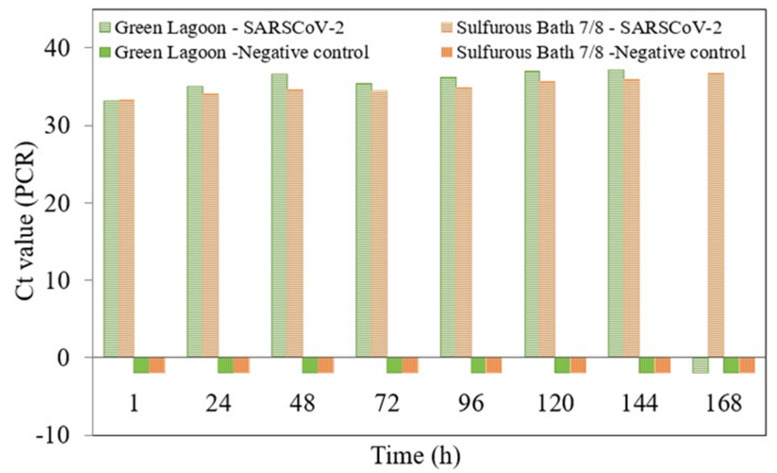
PCR values for sample and control depending on time.

**Figure 3 ijerph-19-05923-f003:**
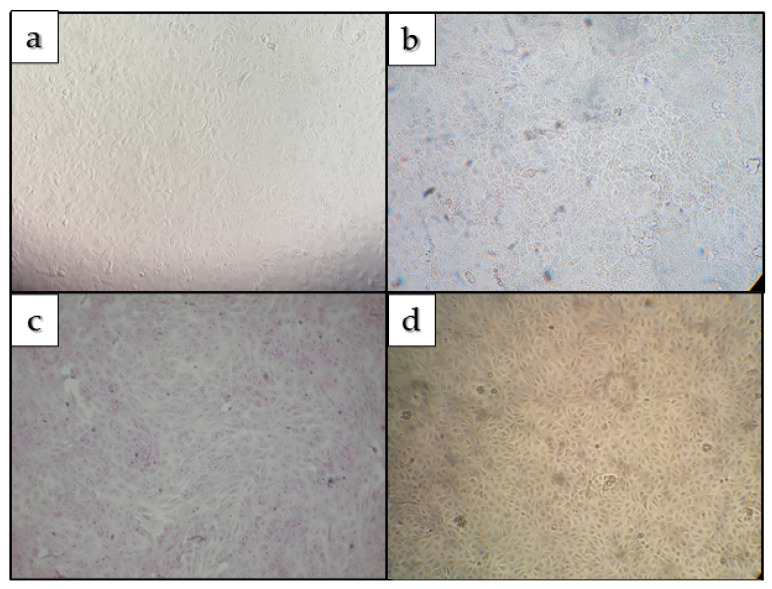
(**a**,**b**): Culture of Vero cells infected with concentrated samples of *Green Lagoon* and Sulfurous *Bath 7/8 Water*, respectively. Figure (**c**,**d**): Uninfected cultures used for control.

**Table 1 ijerph-19-05923-t001:** Physical-chemical composition of Copahue mineromedicinal waters (values expressed in mg·L^−1^). Nd, not detected.

	Chancho’s Lagoon	Green Lagoon	Sulfurous Lagoon	Sulfurous Bath 7/8 Water	Mate Water	Sulfurous Water	Volcan Water
pH	2.3	4.2	5.6	6.6	7.2	7.1	2.8
Conductivity µS cm^−1^	3082	1857	1210	421.1	520	775	44,390
Redox Potentia mV	−65	135	170	−204	−104	−328	427
Temperature	35.4	29.6	54	41.9	60.2	51.5	69
Cl^−^	6.55	1.9	2.7	3.2	1.4	1.2	627.9
SO_4_^2−^	1437	482.6	586.3	33.6	50.9	11.3	23,207.5
CO_2_ gas	nd	nd	nd	19.8	59.4	2.0	nd
H_2_S gas	nd	nd	nd	2.3	nd	0.1	nd
F-	0.31	0.3	0.3	0.2	0.2	0.2	49.9
NO_3_^−^	nd	nd	nd	16.0	2.6	0.8	0.0
HCO_3_^−^	nd	nd	nd	201.3	231.8	463.6	0.0
SH^−^	nd	nd	nd	1.2	nd	nd	nd
Na^+^	52.7	15.4	29.3	33.6	30.2	50.5	1174.5
K^+^	21.21	11.4	19.1	16.5	20.9	27.1	772.3
Sr^2+^	nd	nd	nd	nd	nd	nd	nd
Ca^2+^	76.96	24.5	44.6	30.2	34.6	49.6	752.1
Li+	0.02	0.01	0.02	0.02	0.02	0.05	0.4
NH_4_^+^	134.52	15.4	54.8	nd	5.1	3.3	nd
Mg^2+^	15.55	4.8	11.3	11.2	10.4	25.6	553.2
Fe total	31.2	2.2	2.2	1.2	2.7	0.2	82.00

**Table 2 ijerph-19-05923-t002:** Remaining viral load after three hours of incubation of the waters with an inoculum of SARS-CoV-2 virus.

Group	Water Procedence	Remaining Viral Load Range (RVL) after 3 h of Incubation
A	Green Lagoon	40% > RLV < 44%
B	Mate Water Sulfurous Lagoon	30% > RLV < 36%
C	Sulfurous Bath 7/8 Water Sulfurous Water	20% > RLV < 23%
D	Volcan Water	11% = RLV

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
