# Peer review of "A Preliminary Study of SARS-CoV-2’s Permanence and Potential Infective Capacity in Mineromedicinal Waters of Copahue, Neuquén, Argentina"

_ijerph, 2022, doi:10.3390/ijerph19105923_

Round 1
Reviewer 1 Report
Authors submit the document: A Preliminary Study of SARS-CoV-2’s Permanence and Potential Infective Capacity in Mineromedicinal Waters of Copahue, Neuquén, Argentina. This document is important because many human traditions include treatments in thermal springwater places however, an important question remains: Does SARS-CoV-2 can survive after contact with this type of water?
This document is important however some points need to be clarified.
L93. The temperature, pH, and conductivity of each of the disperse systems were measured periodically in situ. The anion and cation contents of the water in equilibrium with each peloid sample were determined in the laboratory using normalized chemical analyses.
To characterize mineral content of water samples, conductance is most often used as an indicator of total dissolved solids concentration of a solution, however, this is a parameter that is not related to microbicidal activity. ORP is used to monitor redox reactions and disinfection. It would be interested the knowledge of this value.
L113. Two water samples were selected according to acidity, mineralization, and tempera.
Authors used seven samples, there were not too many, I suggest to evaluate all samples instead of only two.
L107. Three aliquots of each sample were infected with a clinical sample of nasopharyngeal swab which had been previously diagnosed positive for SARS-CoV-2 with a viral load of 5754 viral copies/µL.
To evaluate viral survival titers, TCID50% technique provides better results. Usually, this technique requires the use of titrated virus instead of swabs and results are reported in log10 to understand better the viral survival dynamics.
Author Response
Response to Reviewer 1 Comments
Au/ Thank you for the useful comments.
Page 2, line 93. The temperature, pH, and conductivity of each of the disperse systems were measured periodically in situ. The anion and cation contents of the water in equilibrium with each peloid sample were determined in the laboratory using normalized chemical analyses
To characterize mineral content of water samples, conductance is most often used as an indicator of total dissolved solids concentration of a solution, however, this is a parameter that is not related to microbicidal activity. ORP is used to monitor redox reactions and disinfection. It would be interested the knowledge of this value
Au/ The information has been added as suggested. See lines 220-221 and Table 1.
The temperature and the redox potential was measured in situ. The pH and the conductivity of the mineromedicinal waters were measured after centrifugation in the laboratory, and their physical-chemical compositions were determined in accordance with the Standard Methods for the Examination of Water and Wastewater [25].
The ORP values have been included in Table 1.
Page 3, line 113. Two water samples were selected according to acidity, mineralization, and temperature.Authors used seven samples, there were not too many, I suggest to evaluate all samples instead of only two
Au/ The information has been added as suggested. See lines 159-167.
Two water samples were selected according to acidity, mineralization, and temperature, as well as other criteria related to their reutilization, their use in enclosed spaces, and the lack of chlorination systems. These samples were Green Lagoon and Bath 7/8 Water. The first one was used for immersion by large groups of patients and was later collected to be used in bathtubs without having been previously disinfected; the second analyzed source, the pools were used on demand, as immersion baths in enclosed areas without any disinfection treatment. Faced with the potential risk regarding the permanence of viral particles in the environment and in the water, a study was carried out on the potential remaining infective capacity these media.
Page 3, line 107. Three aliquots of each sample were infected with a clinical sample of nasopharyngeal swab which had been previously diagnosed positive for SARS-CoV-2 with a viral load of 5754 viral copies/µL.
To evaluate viral survival titers, TCID50% technique provides better results. Usually, this technique requires the use of titrated virus instead of swabs and results are reported in log10 to understand better the viral survival dynamics.
Au/We agree with the reviewer that one of the methods to assess viral survival titers is the TCID50% technique. This method provides very reliable results. However, the combination of cell culture methods followed by molecular methods for the amplification of SARS-CoV-2 makes it possible to have a system that provides greater detection sensitivity (due to the amplification of the virus in blind passages) and greater specificity. detection using real-time PCR targeting the N gene of SARS-CoV-2. That is why the combination of cellular and molecular methods was chosen to evaluate viral infectivity.
Reviewer 2 Report
Lines 93 -99
In this paragraph appear, disperse systems, peloids and mineromedicinal waters, in the work only water was used, to avoid confusion I suggest modifying it in this way.
The pH and the conductivity of the mineromedicinal waters were measured after centrifugation in the laboratory, and their physical-chemical compositions were determined in accordance with the Standard Methods for the Examination of Water and Wastewater [25]
Line
Line 120:
Where it says: Thermal water; should put: mineromedicinal water
It is the name of the waters given in the title and must be used throughout the text
Line 128
Where it says: PEG 6000. Should put: Polyethylene glycol (PEG 6000)
Line 131
Before: PBS; Indicate the complete name of the buffer
Linea 146
Ct. As this is the first time it appears in the text, the full name must be given. We assume that it refers to the cycling threshold.
Lines 156 a 198
They are not Results. They should be included in the introduction or in materials
Line 216
Where it says: chlorinating; It should put: Chlorination
Line 250
Where it says: chlorinating; It should put: Chlorination
Line 250:
Where it says: mineral-medicinal water; must put mineromedicinal water
Line 268:
Table 2. Explain why Chancho water is not included
Line 285
Where it says: lower CT; You should put: lower Ct
Line 287
Figura 2. Correct Ct on the ordinate axis
Línes 296-297
In the figure unify capital letters (A,B,C,D) and figure captions (a,b,c,d)
Line 297
Were the samples mixed for any reason? Explain it better
Line 308:
Where it says: mineral-medicinal water; must put mineromedicinal water
Linea 313:
Taking into account the different types of minomedicinal waters, I do not think it is possible to state that, in general, “mineromedicinal waters would not be a potential source of viral infection in the exposed population”.
Where it says: mineral-medicinal waters; should put: mineromedicinal
waters used in this study
Line 329
It should be indicated which of the seven watersc
Author Response
Response to Reviewer 2 Comments
Au/ Thank you for the useful comments.
Pages 2-3, lines 93-99. In this paragraph appear, disperse systems, peloids and mineromedicinal waters, in the work only water was used, to avoid confusion I suggest modifying it in this way.
Au/ The pH and the conductivity of the mineromedicinal waters were measured after centrifugation in the laboratory, and their physical-chemical compositions were determined in accordance with the Standard Methods for the Examination of Water and Wastewater [25]
Au/ The information has been added as suggested. See lines 143-146.
Page 3, Line 120. Where it says: Thermal water; should put: mineromedicinal water
It is the name of the waters given in the title and must be used throughout the text
Au/ The information has been added as suggested. See line 81, 119, 142, 144.
Page 3, Line 128. Where it says: PEG 6000. Should put: Polyethylene glycol (PEG 6000)
Au/ The information has been added as suggested. See line 179.
Page 3, Line 131. Before: PBS; Indicate the complete name of the buffer
Au/ The information has been added as suggested. See lines 182.
Page 3, Line 146. Ct. As this is the first time it appears in the text, the full name must be given. We assume that it refers to the cycling threshold.
Au/ The information has been added as suggested. See line 197.
Pages 4-5, lines 156 a 198. They are not Results. They should be included in the introduction or in materials
Au/ The information has been added as suggested. See lines 70-119.
Page 6, Lines 216-217. Where it says: chlorinating; It should put: Chlorination
Au/ The information has been added as suggested. See lines 222-223.
Page 7, Line 250. Where it says: chlorinating; It should put: Chlorination
Au/ The information has been added as suggested. See line 256.
Page 7, Line 250. Where it says: mineral-medicinal water; must put mineromedicinal water
Au/ The information has been added as suggested. See line 256.
Page 7, Line 268. Table 2. Explain why Chancho water is not included
Au/ The information has been added as suggested. See line 274-279.
In table 2 shows six mineral medicinal water, our intention was determined remaining viral load after three hours of incubation of the waters with an inoculum of SARS-CoV-2 virus in the seven spring water. For operational reasons, it was not possible to carry out further tests on the water of the Chancho´s Lagoon, one of the main lagoons of the Copahue hot springs. However, its exclusive outdoor use makes infective capacity determinations less relevant.
Page 8, Line 285. Where it says: lower CT; You should put: lower Ct
Au/ The information has been added as suggested. See line 297.
Page 8, Line 287. Figure 2. Correct Ct on the ordinate axis
Au/ The information has been added as suggested. See line 299-300.
Page 9, Lines 296-297. In the figure unify capital letters (A,B,C,D) and figure captions (a,b,c,d)
Au/ The information has been added as suggested. See lines 307-308.
Page 9, Line 297. Were the samples mixed for any reason? Explain it better
Au/ According to the reviewer, a typo was made in writing the figure legend. It was replaced by: "a and b: Vero cells infected with SARS-CoV-2 in concentrated samples from Green Lagoon and Bath 7/8 Water respectively. c and d: uninfected cultures used for control". See lines 308-310.
Page 9, Line 308. Where it says: mineral-medicinal water; must put mineromedicinal wate
Au/ The information has been added as suggested. See line 320.
Page 9, Linea 313. Taking into account the different types of mineromedicinal waters, I do not think it is possible to state that, in general, “mineromedicinal waters would not be a potential source of viral infection in the exposed population”
Where it says: mineral-medicinal waters; should put: mineromedicinal waters used in this study
Au/ The information has been added as suggested. See line 331-340.
Page 10, Line 329. It should be indicated which of the seven waters
Au/ Copahue mineromedicinal waters from Green Lagoon and Sulfurous Bath 7/8 would not be an adequate matrix for sustaining the viability of SARS-CoV-2. See line 341-342.